# Modifications on the Tetrahydroquinoline Scaffold Targeting a Phenylalanine Cluster on GPER as Antiproliferative Compounds against Renal, Liver and Pancreatic Cancer Cells

**DOI:** 10.3390/ph14010049

**Published:** 2021-01-10

**Authors:** David Méndez-Luna, Loreley Araceli Morelos-Garnica, Juan Benjamín García-Vázquez, Martiniano Bello, Itzia Irene Padilla-Martínez, Manuel Jonathan Fragoso-Vázquez, Alfonso Dueñas González, Nuria De Pedro, José Antonio Gómez-Vidal, Humberto Lubriel Mendoza-Figueroa, José Correa-Basurto

**Affiliations:** 1Laboratorio de Diseño y Desarrollo de Nuevos Fármacos e Innovación Biotecnológica (Laboratory for the Design and Development of New Drugs and Biotechnological Innovation), Escuela Superior de Medicina, Instituto Politécnico Nacional, Plan de San Luis y Díaz Mirón, s/n, Col. Casco de Santo Tomas, Ciudad de México 11340, Mexico; meld8909@gmail.com (D.M.-L.); morelos.loreley@gmail.com (L.A.M.-G.); bellomartini@gmail.com (M.B.); mzalubriel@hotmail.com (H.L.M.-F.); 2Departamento de Fisiología, Escuela Nacional de Ciencias Biológicas, Instituto Politécnico Nacional Zacatenco, Av. Wilfrido Massieu 399, Col. Nueva Industrial Vallejo, Alcaldía Gustavo A. Madero, Ciudad de México 07738, Mexico; 3Laboratorio de Química Supramolecular y Nanociencias, Unidad Profesional Interdisciplinaria de Biotecnología, Instituto Politécnico Nacional, Av. Acueducto s/n., Barrio La Laguna Ticomán, Ciudad de México 07340, Mexico; ipadillamar@ipn.mx; 4Departamento de Química Orgánica, Escuela Nacional de Ciencias Biológicas, Instituto Politécnico Nacional Prolongación de Carpio y Plan de Ayala S/N. Col. Casco de Santo Tomas, Ciudad de México 11340, Mexico; mjfragosov@hotmail.com; 5Genomic Medicine and Environmental Toxicology, Biomedical Research Institute, UNAM, National Cancer Institute, Av San Fernando 22, Tlalpan, Mexico City 14080, Mexico; alfonso_duenasg@yahoo.com; 6Fundación MEDINA, Parque Tecnológico Ciencias de la Salud, Avenida del Conocimiento 34, 18016 Granada, Spain; ndepedro@lifelength.com; 7Facultad de Farmacia, Departamento de Química Farmacéutica y Orgánica, Universidad de Granada, 18071 Granada, Spain; jagvidal@gmail.com

**Keywords:** GPER, docking, molecular dynamics simulations, Suzuki–Miyaura cross-coupling, tetrahydroquinoline scaffold, antiproliferative

## Abstract

The implementation of chemo- and bioinformatics tools is a crucial step in the design of structure-based drugs, enabling the identification of more specific and effective molecules against cancer without side effects. In this study, three new compounds were designed and synthesized with suitable absorption, distribution, metabolism, excretion and toxicity (ADME-tox) properties and high affinity for the G protein-coupled estrogen receptor (GPER) binding site by in silico methods, which correlated with the growth inhibitory activity tested in a cluster of cancer cell lines. Docking and molecular dynamics (MD) simulations accompanied by a molecular mechanics/generalized Born surface area (MMGBSA) approach yielded the binding modes and energetic features of the proposed compounds on GPER. These in silico studies showed that the compounds reached the GPER binding site, establishing interactions with a phenylalanine cluster (F206, F208 and F278) required for GPER molecular recognition of its agonist and antagonist ligands. Finally, a 3-(4,5-dimethylthiazol-2-yl)2,5-diphenyltetrazolium bromide (MTT) assay showed growth inhibitory activity of compounds **4**, **5** and **7** in three different cancer cell lines—MIA Paca-2, RCC4-VA and Hep G2—at micromolar concentrations. These new molecules with specific chemical modifications of the GPER pharmacophore open up the possibility of generating new compounds capable of reaching the GPER binding site with potential growth inhibitory activities against nonconventional GPER cell models.

## 1. Introduction

It is widely known that steroid hormones play a crucial role in carcinogenesis and cancer progression [1]. Among this group of hormones, estrogen regulates several physiological and disease processes in humans at a transcriptional level, known as the genomic mechanism [2]. Additionally, there is a nongenomic level that involves rapid intracellular signaling, mediated through second messengers that are initiated by the activation of membrane receptors, specifically G protein-coupled receptors (GPCRs) [3]. Recently, a new estrogen membrane receptor was revealed as a pharmacological target in breast cancer, encouraging researchers to find new ligands against breast cancer [3,4]. Originally, this receptor was named as GPR30 but is now referred to as G protein-coupled estrogen receptor (GPER) by the International Union of Basic and Clinical Pharmacology (IUPHAR; www.iuphar.org/). GPER belongs to the superfamily of seven-transmembrane-spanning GPCRs, which are the most important group of pharmacological targets in the pharmaceutical market [5,6]. Specifically, GPER belongs to the class A rhodopsin-like GPCR family; its primary transduction mechanism is led through coupling with Gα_i/o_ proteins and its secondary through the Gα_s_ protein. GPER activates a network of transduction pathways, such as formation of the epidermal growth factor receptor (EGFR), the mitogen-activated protein kinases (MAPK), increment of the intracellular cyclic AMP (cAMP) and calcium mobilization [7,8,9]. Dihydroquinoline derivative G1, the first synthetic agonist ligand targeting GPER [10] that does not show affinity for estrogen receptors (ERα and ERβ), contains a structural scaffold homologous to other GPER antagonists (G15 and G36), and they differ only by the chemical moieties that could be responsible for their biological responses as either agonists or antagonists [10,11,12,13].

As a promising agent for use in cancer therapy, GPER could be a biological target in a wide range of cancer cells, producing positive results as quantified by their growth inhibitory activity, including renal [14], liver [15] and pancreatic [16] cancers. It is important to note that the biological activity of GPER is tissue-dependent as well as whether the cells are healthy or cancerous and the ability of tissues to express GPER [17,18]. Recent structural insights regarding the agonist/antagonist activities of GPER through in silico methods to explain how and why the activation of this receptor occurs have been applied to find new and more efficient ligands that target GPER [19,20]. In a previous study, docking and molecular dynamics (MD) simulations were combined with a molecular mechanics/generalized Born surface area (MMGBSA) approach to reveal the structural and energetic basis of the molecular recognition between agonists or antagonists on the GPER binding site. This study elucidated key residues of GPER involved in molecular recognition, including a phenylalanine cluster (F206, F208 and F278) and a polar residue (N310). The latter is considered to be a highly important residue involved in the activation or inactivation mechanism of GPER [21].

In accordance with the knowledge of key amino acid residues involved in the molecular recognition that dictates the binding mode of agonists and antagonists in the active site of GPER, we focused our efforts on proposing strategic chemical modifications on the pharmacophore that would improve molecular recognition through additional interactions and, in turn, explore additional cavities that would increase selectivity and affinity at the GPER binding site. First, we substituted the bromine atom with a phenyl ring, which was in turn substituted with electron withdrawing (*m*-NO_2_) (compound **4**) or electron donating (*m*-OCH_3_) (compound **5**) groups. The second chemical modification was to obtain a molecule with a tert-butyl group off the piperidine ring (compound **7**), seeking to increase the hydrophobic environment that can establish interactions with the aforementioned cluster of aromatic residues. Thus, we started with a G1 analog previously prepared by our research group [22], which is subsequently referred to as G1-PABA. The chemical synthesis of G1-PABA involved modification of the *p*-amino acetophenone of G1 using *p*-aminobenzoic acid, which has shown inhibitory activity against breast cancer cells [22,23]. Once G1-PABA was synthetized, compounds **4**, **5** and **7** were synthesized and tested in renal, liver and pancreatic cancer cells. The results showed growth inhibitory activities (half maximal inhibitory concentration (IC_50_) values) less than 50 µM. In addition, the structural and energetic properties of **4**, **5** and **7** in the GPER binding site were explored by combining docking and MD simulations with a MMGBSA approach, enabling us to correlate the experimental and theoretical results.

## 2. Results and Discussion

### 2.1. Docking Calculations

To analyze the possible affinity and binding posture of the designed ligands in the GPER binding site, docking studies were performed, and it was found that the designed ligands were capable of reaching cavities in addition to those observed in previous results [20,21]. As we reported previously, the molecular recognition conducted by GPER involves essential residues, mainly aromatic in nature, comprising a phenylalanine cluster (F206, F208 and F278) and a polar residue (N310). Accordingly, in this study, in which great chemical modifications of the chemical scaffold were explored (hindrance properties, electron donation and electron withdrawing as well as hydrophobic effects), either agonist or antagonist GPER ligands examined in silico showed binding modes on GPER that were equivalent to our previous findings [20,21]. Indeed, ligand **4** (*m*-NO_2_ phenyl group) appeared to be in close contact with the phenylalanine cluster due to the ionic–π interaction exerted by the electroattractant NO_2_ group facing the π electron density of the phenyl groups in the phenylalanine cluster. This property was reflected in the better binding energy of **4** (−8.46 Kcal/mol) compared to **5** (−8.36 Kcal/mol) due to the latter having an electron-donating OCH_3_ group (Table 1). Ligand **7** showed a binding mode in which the Boc (di-tert-butyl dicarbonate) moiety was in close contact with N310 (less than 5 Å). The other important feature of **7** was that the orientation of its binding mode overlapped with the binding mode of **5**, sharing almost the same contacts with the residues involved in the binding site (Table 1).

It is important to note that the pivotal residues (phenylalanine cluster and N310) were not found at the same time to be involved in nonbonding interactions with the ligands during the first docking recognition study, although it is well known that GPER contains more than one binding site formed by residues of the same chemical nature as an orthosteric binding site [20]. The resulting binding pose of the ligands obtained from the docking on GPER (for an atomistic depiction please see the Appendix A) were employed as input coordinates for MD simulations combined with energetic calculations using the MMGBSA approach. In addition, the ligands showed acceptable physicochemical and toxicological–biological properties (absorption, distribution, metabolism, excretion and toxicity (ADME-tox)) according to Lipinski’s rules, which are the parameters required for new drug design (see Table 2). Toxicological–biological evaluations showed no effect on any property analyzed.

### 2.2. Molecular Dynamics Simulations

#### 2.2.1. Evaluation of the Equilibrium of Membrane-Embedded MD GPER–Ligand Complexes

Docking studies demonstrated that the compounds studied established a different map of contacts to those previously identified due to hindrance effects [20,21] involving interactions with the phenylalanine cluster (F206, F208 and F278) and a polar residue (N310) (data not shown). However, to provide more consistent information about the prevalence of the interactions observed through docking calculations as well as to provide conformational information, the GPER–ligand complexes were membrane-embedded and submitted through 100 ns long MD simulations, accompanied by binding free energy calculations using the MMGBSA approach to obtain a system with better physiological conditions than rigid strategies.

Before performing any analysis of the structural and energetic values of the simulated membrane-embedded GPER–ligand complexes, we evaluated the MD simulation time required to perform this analysis once the equilibration stages were reached (Figure 1). Therefore, three geometrical values were evaluated to determine whether the systems reached equilibrium: area per lipid of the 1-palmitoyl-2-oleoyl-sn-glycero-3-phosphorylcholine (POPC) membrane; root mean square deviation (RMSD) of heavy atoms; and the radius of gyration (Rg) of the GPER with respect to the starting conformation. Figure 1A shows the MD simulation time evolution of the area per lipid for the three GPER–ligand complexes. It was observed that for the first 30 ns of each MD simulation, the area per lipid values were high but then decreased and converged at 60 ns to values of 61.03 ± 0.26, 61.34 ± 0.30 and 60.76 ± 0.32 Å for GPER-**4**, GPER-**5** and GPER-**7**, respectively. RMSD analysis showed that the three GPER–ligand complexes reached equilibrium at similar MD simulation times (approximately 50 ns) with RMSD values of 2.6 ± 0.12, 2.2 ± 0.17 and 2.5 ± 0.11 Å for GPER-**4**, GPER-**5** and GPER-**7**, respectively (see Figure 1B). Similarly, the Rg analysis illustrated that the three systems reached equilibrium at 60 ns with Rg values of 26.5 ± 0.1, 26.4 ± 0.13 and 26.5 ± 0.12 Å for GPER-**4**, GPER-**5** and GPER-**7**, respectively (see Figure 1C). Based on these observations, it was evident that the three systems reached equilibrium at 60 ns, from which root mean square fluctuation (RMSF) calculations and subsequent analyses were performed. The RMSF analysis over the equilibrated simulations (the last 40 ns of the MD production runs) revealed that the molecular flexibility of the three systems were similar (see Figure 1D). The loop (residues 20–50) that flanked the receptor binding site was the region that exhibited the highest conformational mobility for the three systems, whereas loop 5 (residues 245–251) exhibited the highest flexibility for GPER-**7** only (see Figure 1D).

#### 2.2.2. Clustering Analysis

Clustering analysis over the equilibrated time was carried out for the GPER-**4**, GPER-**5** and GPER-**7** complexes to detect all conformational states of each complex. Clustering analysis using a cut-off of 2.0 Å demonstrated that GPER-**4**, GPER-**5** and GPER-**7** complexes were characterized by one, three and five clusters of conformers, respectively, with 100%, 73% and 56% of the clusters of each GPER-ligand complex being represented by the first cluster, respectively. This analysis allowed us to obtain the most populated conformation, which was useful in evaluating the map of interactions between the ligands and GPER.

#### 2.2.3. Principal Component Analysis (PCA)

The total fluctuation of the GPER-**4**, GPER-**5** and GPER-**7** complexes was evaluated using Cartesian principal component analysis (cPCA) (see Methods). This analysis revealed that the total motion of the GPER-**4**, GPER-**5** and GPER-**7** complexes was dispersed over 3375 eigenvectors; however, the total MD simulations were described by 15 eigenvectors. Figure 2 shows the first 15 collective modes (Figure 2A) for the three systems and their individual percentages (Figure 2B). The collective modes captured 69.5%, 69.4% and 75.3% of the total motion for the GPER-**4**, GPER-**5** and GPER-**7** complexes, respectively. The projection of GPER-**4**, GPER-**5** and GPER-**7** onto eigenvectors 1 and 2 (PC1 vs. PC2) showed that the fluctuation of the three systems was confined within these two eigenvectors. However, GPER-**7** covered a larger region of phase space along the PC1 and PC2 plane than GPER-**4** and GPER-**5**, and this latter complex covered a larger region of phase space along the PC1 plane than GPER-**4**. The clusters of stable states were more defined in GPER-**4** than GPER-**7** and GPER-**5**. Quantification of the total flexibility among the different protein–ligand complexes, shown in Figure 2C, was obtained by analyzing the trace of the diagonalized covariance matrix of the backbone atomic positional fluctuations. Based on this parameter, the values for GPER-**7** (12.71 nm^2^), GPER-**4** (9.35 nm^2^) and GPER-**5** (9.94 nm^2^) revealed differences in conformational flexibility in the order of GPER-**7** > GPER-**5** > GPER-**4**. This order was in line with the conformational complexity identified through the cluster analysis.

#### 2.2.4. Structural Analysis of the Empty and Bound GPER Systems

Structural details of a map of the interactions for the GPER–ligand complexes were analyzed considering the most populated conformation obtained through cluster analysis (Figure 3). A comparison of the map of interactions of the three complexes showed that **7** (Figure 3B) and **4** (Figure 3C) shared contact with five of the same residues: Gln53, Leu59, Arg122, Met133 and Leu137. Ligands **7** and **5** also shared contacts with Gln53, Leu59, Glu275, Arg122 and Gly58, and **5** and **4** shared contacts with Gln53, Leu59, Leu119, His120 and Arg122. However, only three residues (Gln53, Leu59 and Arg122) were shared by all three compounds, indicating that small modifications in the pharmacophore group can modify the map of interactions.

Overall, this analysis revealed that the three compounds were mostly stabilized by hydrophobic residues. Residues belonging to the phenylalanine cluster (F206, F208 and F278) were identified as important stabilizers of the complexes with GPER [20,21]. Among the three complexes, only **4** (F206 and F208) and **5** (F278) were able to establish contact with some members of this cluster. With respect to polar interactions, only **7** and **4** were able to establish hydrogen bonds and salt bridges through their polar moieties at the GPER binding site. Ligand **7** formed a hydrogen bond with Gln53 and a salt bridge with Arg122 through its carboxyl group (Figure 3B), whereas **4** was stabilized by two hydrogen bonds through its carboxyl group with His120 and its NO_2_ group with Cys207 (Figure 3C). The latter polar interaction has been reported to be responsible for inducing the activated state [20]. Interestingly, the lack of a bromine (Br) atom in these compounds means that the polar interactions observed in G1 and G15 through Asn307 or Asn310, respectively [21], were not present. Instead, these two residues appeared in GPER-**7** and GPER-**5**, forming polar interactions through backbone atoms.

#### 2.2.5. Absolute Binding Free Energy Calculations

The absolute binding free energy (ΔG_bind_) was calculated to measure the energetic contributions in terms of noncovalent energies at the binding site of the GPER-**4**, GPER-**5** and GPER-**7** complexes (see Methods). This analysis revealed that the Δ*G_bind_* values for the three GPER–ligand systems, considering the entropic component, were thermodynamically favorable for all complexes (Table 3). The main energetic contribution to Δ*G_bind_* was guided by the nonpolar contributions (Δ*E_nonpolar_ =* Δ*E_vwd_ +* Δ*G_npol,sol_*), whereas the polar contributions (Δ*E_polar_ =* Δ*E_ele_ +* Δ*G_pol,sol_*) displayed a thermodynamically unfavorable performance that contrasted with the binding process for the three complexes. Based on the entropic values (Table 3), it was observed that the three complexes exhibited a significant conformational reduction upon complex formation, notably impacting the Δ*G_bind_* value, suggesting that this molecular recognition was guided through an unfavorable entropic component. The GPER-**4** system exhibited a more energetically favorable Δ*G_bind_* than the GPER-**7** and GPER-**5** complexes, suggesting that the NO_2_-containing compound established a more energetic map of interactions into the GPER binding site than the other two compounds.

### 2.3. Antiproliferative Assays

GPER plays a pivotal role in triggering cancer development, and most studies in the literature have focused on its overexpression in breast cancer. The current findings relate to the properties of GPER, particularly that its activity depends specifically on the expression tissue and the cognate ligands (either agonist or antagonist) that activate or inactivate it. Thus, new possibilities have been generated in the evaluation of new compounds targeting GPER in cancer cells, unlike classical models of activity evaluation [24,25,26]. However, there are other types of cancer in which GPER is present, and it could be an interesting biological target in some other cancers, such as renal [14], liver [15] and pancreatic [16]. The evaluation of growth inhibition of the synthetized compounds was conducted using a 3-(4,5-dimethylthiazol-2-yl)2,5-diphenyltetrazolium bromide (MTT) assay in selected cancer cell lines. RCC4-VA, RCC4-VHL, MIA Paca-2 and Hep G2 were used to investigate the possible antiproliferative effects of these three compounds as these are diverse cancer cells that all have GPER in common. As shown in Table 4, MIA Paca-2 was the most sensitive cancer cell line to the compounds tested. In this cell line, all tested compounds had an antiproliferative effect in a concentration-dependent manner. The most effective ligand in the MIA Paca-2 cell line was **7**, which has a Boc protecting group on the amine of the piperidine ring of the pharmacophore of GPER (see Figure 4). In this cell line, the expression pattern of estrogen receptors is known, and antiestrogen therapy is used to block ER activity in order to treat the cancer [27,28].

The GPER signaling pathway was studied and its role in both healthy and cancerous processes were examined, particularly with regard to the very important role in the tumor microenvironment for renal, liver and pancreatic cancers, which could be affected by the synthetized compounds. Interestingly, only two compounds showed an effect in mediating growth inhibition in the RCC4-VA cell line (**7** and **5**), but with higher concentrations than those assayed in MIA Paca-2 cells (see Figure 5). In this cell line, it is important to note that the trend of **4**, which showed a non-dose-dependent behavior (evident at 6.3 µM concentration), might be due to intrinsic properties of the cell to adapt to increased concentrations of the tested compounds. In this case, the cell could be increasing the expression of proteins involved into externalize the drugs (e.g., P-glycoprotein (PGP) ATPase) or another mechanism, which conferred to the cell the capability to not respond to the treatments assayed. This correlated with the result yielded by **7** assayed at the highest concentration, although the result was slightly lower than the previous concentration (25 µM). This pharmacological phenomenon is named as a biphasic pattern and has been observed for other cancer cells (https://www.ncbi.nlm.nih.gov/pmc/articles/PMC4560104/). In contrast, in the RCC4-VHL cell line, all compounds showed activity at the concentrations assayed (data not shown). This result is promising due to the molecular differences by which these cell lines are employed, noting that VHL/HIF deregulated signaling and the distinctive molecular feature of RCC4-VA cells responsible for the high rate of tumors characteristic of Von Hippel-Lindau syndrome [29,30]. Finally, in Hep G2 (human liver carcinoma) cells, only compound **5** showed activity (Figure 6) with almost the same concentration found for RCC4-VA. This result might unmask an axis that involves the participation of GPER/HIF-1α/vascular endothelial growth factor (VEGF) as has been reported elsewhere [31]. Additionally, all cell models tested have been implicated in processes in which the participation of molecular mediators under hypoxic conditions are initiated once GPER activation has occurred [32,33]. These findings suggest that GPER is a potential pharmacological target to develop new therapeutic strategies for cancer.

## 3. Materials and Methods

### 3.1. Docking, Physicochemical and ADME-Tox Properties Analysis

The potential interaction between the designed compounds and GPER was analyzed using molecular docking studies. First, the two-dimensional (2D) chemical structures of the ligands were drawn with the ChemBioDraw Ultra 12.0 program [34]. The Z-matrices of the structures were generated using the Gauss View 5.0 program [35] to assign the correct interatomic distance and hydrogen types. This step was undertaken through energetic and geometric minimization under an AM1 semiempirical method that employed the quantum chemistry package Gaussian 09 [36]. The GPER three-dimensional (3D) model previously generated and validated by our research group was used in the molecular docking simulations [20]. These simulations were carried out using the AutoDock 4.2.6 software [37]. Polar hydrogen atoms were added to the polar atoms of the protein, and Kollman charges were assigned. Gasteiger charges were assigned for ligands, and a grid-based procedure was used to generate the affinity maps delimiting the protein binding site with the following grid box parameters: XYZ of 60 Å^3^ over the Cα atom of the N310 residue (focused procedure) with a grid spacing of 0.375 Å^3^. Scoring sampling used the Lamarckian genetic algorithm considering the following as search parameters: a randomized initial population of 100 individuals and a maximum number of energy evaluations of 1 × 10^7^ cycles. The ligand–receptor interactions were analyzed with PyMOL v0.99 graphical viewer [38]. The physicochemical and ADME-tox properties were analyzed using the Molinspiration server [39] and DataWarrior software [40], respectively.

### 3.2. Embedding of the GPER–Ligand Complexes into the Membrane

Orientation of the GPER-**4**, GPER-**5** and GPER-**7** complexes with respect to the POPC membrane was performed using the Orientations of Proteins in Membranes (OPM) server [41]. Rectangular pre-equilibrated POPC bilayers were built for each complex using the membrane builder tool CHARMM [42] with dimensions 110.437 × 110.437 × 122.485 Å (xyz). GPER–ligand complexes were embedded into the POPC bilayer using the replacement method in CHARMM [43,44]. POPC bilayers constituted approximately 314 POPC phospholipids with 154 and 160 on the outer and the inner leaflets, respectively, and they were solvated using the TIP3 water model and neutralized with an ionic strength of 0.15 M by NaCl using the solvation and autoionize module [43].

### 3.3. MD Simulations of the Membrane-Embedded GPER–Ligand Complexes

The MD simulations were performed with pmemd.cuda AMBER 12 executable [45]. Topologies for the GPER–ligand complexes were built using the Leap module, employing the ff99SB, Lipid11 and GAFF force fields [46,47]. These complexes were minimized and equilibrated with the sander module through the following steps. The complexes were energy minimized through 10,000 steps with position restraints on the receptor and lipids to allow relaxation of the water molecules. Complexes were slowly heated under constant temperature and constant volume (NVT) ensemble through two sequential runs from 0 to 300 K for 1 ns, maintaining the receptor atom and lipid restraints. Afterwards, a 1 ns simulation under the NPT ensemble at 300 K and 1 bar pressure with the heavy atoms restrained was carried out to equilibrate the system, followed by 1 ns of equilibration with the fully unrestricted system. The equilibration runs were followed by 100 ns MD simulations, which were run for each system without position restraints under periodic boundary conditions (PBCs) using a NPT ensemble at 300 K and 1 bar pressure. The electrostatic interactions were computed by the particle mesh Ewald (PME) method [48]. A 10 Å cut-off was used for the van der Waals interactions, and the SHAKE algorithm was used to constrict the bonds between the heavy atoms and hydrogen atoms [49]. The temperature was conserved using Langevin dynamics, and pressure was controlled using a semi-isotropic constant surface tension to preserve a specific area per lipid. The pressure was maintained at 1 bar, and the pressure coupling constant was set to 1 ps with a collision frequency of 1 ps^−1^. The time step was set to 2 fs, and the coordinates were saved for analysis every 1 ps.

### 3.4. Analysis of MD Simulations

The RMSD and Rg were calculated only for the α-carbon atoms and the receptor, respectively, after removing the overall translational and rotational motion. The area per lipid was calculated using the following equation: area per lipid = (box X length × box Y length)/number of phospholipids per layer. The most populated conformers, which are by definition the most stabilized systems present in a physiological context, were obtained through a RMSD conformational clustering analysis using the g_cluster command with the GROMOS algorithm and a cut-off of 0.25 nm [50] to evaluate the conformational features linked to the most stable GPER–ligand complexes. Images and structural representations were prepared using PyMOL v0.99 [38].

### 3.5. Principal Component Analysis

Principal component analysis, the essential dynamics (ED) method [51] and the g_covar and g_anaeig GROMACS 4.5.3 package utilities [52,53] were implemented to predict the collective motion of atoms. The symmetric matrix was diagonalized to form a covariance matrix, which was used to obtain a set of eigenvectors and eigenvalues. The essential subspace relevant to the GPER–ligand systems was constructed by projecting the PCA along the most significant eigenvectors.

### 3.6. Calculation of Absolute Binding Free Energies

The binding free energy was computed for the GPER–ligand complexes using the MMGBSA method and the single trajectory approach [54,55,56,57,58,59,60,61]. To this end, the water molecules and counterions were removed from the trajectory, and the energy computations were performed by selecting 400 snapshots at time intervals of 100 ps from the last 40 ns of the simulation run for each system. The binding free energy (Δ*G_bind_*) of each GPER–ligand complex was calculated as follows:(1)$$∆G_{bind}=G^{complex}-G^{receptor}-G^{ligand}$$
(2)$$=∆E_{MM}+∆G_{GB}+∆G_{SA}-T∆S$$
where Δ*E_MM_* corresponds to the gas-phase interaction energy for the protein–ligand complex, which includes the van der Waals (Δ*E_vdw_*) and the electrostatic (Δ*E_ele_*) interaction energies; Δ*G_GB_* and Δ*G_SA_* are the electrostatic and nonpolar contributions to desolvation upon ligand binding, respectively; and −*T*Δ*S* is the entropy contribution, which is a result of the structural changes in the degrees of freedom between the free and bound partners forming the protein–ligand complex. Because −*T*Δ*S* was considered in the computation of the binding free energy calculations, the values we obtained using the MMGBSA approach could be named as absolute binding free energies, Δ*G_bind_*.

### 3.7. Calculation of Entropy Contributions

The entropy contribution was evaluated for each complex using the MMPBSA.py module implemented in Amber 12. Due to the large memory required for this calculation, only 20 snapshots were selected for each GPER–ligand complex at time intervals of 2 ns from the last 40 ns of each simulation run.

### 3.8. Chemical Synthesis

All reagent-grade chemicals were obtained from commercial suppliers and were used without further purification. Reactions were monitored by thin-layer chromatography (TLC) on aluminum-backed sheets with silica gel 60 GF_254_ (HX805651) and a fluorescent indicator (visualized with UV light of 254 nm). Flash chromatography was performed using silica gel 60 (230–400 mesh). Melting points (mp) were determined on an Electrothermal IA 91000 apparatus (Electrothermal, Bibby Scientific, Staffordshire, ST15 OSA, UK) and were uncorrected. ^1^H and ^13^C NMR spectra were obtained on a Varian Mercury 300 MHz spectrometer or a Bruker Avance III 750 MHz spectrometer using deuterated dimethyl sulfoxide (DMSO-*d*_6_) or deuterated chloroform (CDCl_3_) as the solvent and tetramethylsilane (TMS) as the internal standard. Chemical shifts (δ) are reported in ppm downfield from the internal standard, and coupling constants are reported in Hertz (Hz) (Appendix A). Electrospray ionization high-resolution mass spectrometry positive mode (ESI-HRMS) was performed with an Agilent 6545 QTOF LC/MS instrument (Agilent Technologies, Santa Clara, CA, USA).

#### 3.8.1. Synthesis of (3a*S*, 4*R*, 9b*R*)-4-(6-Bromobenzo[d][1,3]dioxol-5-yl)-3a,4,5,9b-tetrahydro-3*H*-cyclopenta[c]quinoline-8-carboxylic acid, G1-PABA **1**

G1-PABA (**1**) and the tert-butyl derivative (compound **7**) were obtained following the previous method reported by our research group [22,23].

#### 3.8.2. Synthesis of (3a*S*, 4*R*, 9b*R*)-4-(6-(3-Nitrophenyl)benzo[d][1,3]dioxol-5-yl)-3a,4,5,9b-tetrahydro-3*H*-cyclopenta[c]quinoline-8-carboxylic acid, **4**

Compound **4** was obtained by following the previously reported method [62]. G1-PABA (0.020 g, 0.048 mmol), *m*-nitro phenylboronic acid (0.0096 g, 0.057 mmol), PEG2000 (0.060 g, 0.10 mmol) and Pd(AcO)_2_ (0.004 g, 0.017 mmol) were added to a solution of K_2_CO_3_ (0.0266 g, 0.19 mmol) and MeOH/H_2_O (7 mL) in a 1:1 mixture with constant stirring (see Scheme 1). The reaction mixture was heated to reflux at 70 °C for 5 h under N_2_ atmosphere. The mixture was then enabled to cool to room temperature, and the resulting mixture was extracted with EtOAc (3 × 25 mL). The organic phase was washed with distilled water and brine, dried over anhydrous sodium sulfate (Na_2_SO_4_) and concentrated under reduced pressure. The residue was purified by flash chromatography using a mixture of hexane/EtOAc (7:3) as the eluent, affording 55% yield of compound **4** as a clay solid, R*_f_* = 0.42 (hexane/EtOAc, 7:3), mp = 184–185 °C. HPLC purity = 97.19 %. ^1^H NMR (750 MHz, CDCl_3_) δ 8.22 (d, 1H, 7.5 Hz, H-4′′), 8.14 (bs, 1H, H-2′′), 7.68 (d, 1H, *J* = 7.5 Hz, H-7), 7.67 (s, 1H, H-9), 7.60 (d, 1H, *J* = 7.5 Hz, H6′′), 7.57 (t, 1H, *J* = 7.5 Hz, H5′′), 7.26 (s, 1H, H-7′), 7.23 (s, 1H, H-4′), 6.68 (s, 1H, NH), 6.56 (d, 1H, *J* = 7.5 Hz, H-6), 6.06 and 6.05 (AB, 2H, H-2′), 5.88 (bm, 1H, H-1), 5.69 (bm, 1H, H-2), 4.29 (d, 1H, *J* = 3.7 Hz, H-4), 4.22 (d, 1H, *J* = 6.0 Hz, H-9b), 2.67 (m, 1H, H-3a), 1.71 (m, 1H, H-3 down), 1.44 (m, 1H, H-3 up). ^13^C NMR (187.5 MHz, CDCl_3_) δ 167.9 (CO_2_H), 150.4 (C-7a’), 148.3 (C-3′’), 148.2 (C-3′), 146.9 (C-5a), 142.5 (C-1′′), 135.6 (C-6′′), 133.9 (C-2), 133.1 (C-6′), 132.6 (C-9), 132.0 (C-1), 130.6 (C-4′), 129.6 (C-5′′), 129.5 (C-2′′), 124.8 (C-7), 124.4 (C-4′′), 122.6 (C-9a), 119.4 (C-8), 115.3 (C-6), 110.4 (C-4′), 107.1 (C-7′), 101.8 (C-2′), 53.6 (C-4), 45.6 (C-9b), 39.0 (C-3a), 38.9 (C-3). HRMS (ESI) calculated for ([C_26_H_20_N_2_O_6_] + H)^+^: 457.1400; found: 457.13824.

#### 3.8.3. Synthesis of (3a*S*, 4*R*, 9b*R*)-4-(6-(3-Methoxyphenyl)benzo[d][1,3]dioxol-5-yl)-3a,4,5,9b-tetrahydro-3*H*-cyclopenta[c]quinoline-8-carboxylic acid, **5**

Compound **5** was obtained through a Suzuki–Miyaura cross-coupling reaction [63]. A solution of K_2_CO_3_ (0.0266 g, 0.19 mmol) in dimethylformamide (DMF, 1.5 mL) and water (1.5 mL) was stirred for 15 min under N_2_ atmosphere. Subsequently, G1-PABA (0.020 g, 0.048 mmol), methoxy-nitro phenyl boronic acid (0.0096, 0.057 mmol), triphenylphosphine (0.0015 g, 0.0057 mmol) and Pd(AcO)_2_ (0.0004 g, 0.017 mmol) were added. The reaction mixture was heated to 100 °C for 24 h (see Scheme 2). Once the reaction was complete, it was allowed to cool to room temperature, and the resulting mixture was extracted with EtOAc (3 × 25 mL). The organic layer was washed with distilled water and brine, dried over anhydrous Na_2_SO_4_ and concentrated under reduced pressure. The obtained residue was purified by flash chromatography with a mobile phase consisting of hexane/EtOAc (8:2), affording a 13% yield of compound **5** as a gray solid, R*_f_* = 0.5 (hexane/EtOAc 8:2), mp = 200 °C. HPLC purity = 87.31 %. ^1^H NMR (300 MHz, CDCl_3_) δ 7.69 (d, 1H, *J* = 7.1 Hz, H-7), 7.68 (s, 1H, H-9), 7.31 (t, 1H, *J* = 8.8 Hz, H-5′′), 7.18 (s, 1H, H-7′), 6.88 (dd, 1H, *J* = 7.9, 2.9 Hz, H-6′′), 6.82 (d, 1H, *J* = 7.6 Hz, H-4′′), 6.78 (d, 1H, *J* = 1.7 Hz, H-2′′), 6.71 (s, 1H, H-4′), 6.54 (d, 1H, *J* = 8.8 Hz, H-6), 6.02 (m, 2H, H-2′), 5.90 (m, 1H, H-1), 5.69 (m, 1H, H-2), 4.70 (d, 1H, *J* = 3 Hz, H-4), 3.81 (s, 3H, OMe), 3.81 (dd, 1H, *J* = 17.6, 8.8 Hz, H-9b), 2.80 (dt, 1H, *J* = 8.3, 3 Hz, H-3a), 2.64 (dd, *J* = 15.2, 8.3 Hz, H-3 down), 1.97 (m, 1H, H-3 up). ^13^C NMR (75 MHz, CDCl_3_) δ 171.6 (CO_2_H), 159.3 (C-1′′), 150.6 (C-5a), 147.4 (C-3′a), 142.0 (C-7′a), 146.2 (C-1′′), 135.1 (C-5′), 132.6 (C-9a), 124.7 (C-6′), 118.8 (C-8), 101.2 (C-2′), 131.8 (C-7), 129.3 (C-9), 128.0 (C-5′′), 106.5 (C-7′), 112.3 (C-6′′), 121.7 (C-4′′), 115.2 (C-2′′), 110.3 (C-4′), 114.9 (C-6), 133.7 (C-1), 130.6 (C-2), 53.2 (C-4), 55.2 (OMe), 45.4 (C-9b), 44.5 (C-3a), 31.9 (C-3). HRMS (ESI) calculated for ([C_27_H_23_NO_5_] + H)^+^: 442.1654; found: 442.1646.

#### 3.8.4. Synthesis of (3a*S*, 4*R*, 9b*R*)-4-(6-Bromobenzo[d][1,3]dioxol-5-yl)-5-(tert-butoxycarbonyl)-3a,4,5,9b-tetrahydro-3*H*-cyclopenta[c]quinoline-8-carboxylic acid, **7**

To a cold solution of G1-PABA in DMF (3.0 mL), triethylamine (336 μL, 2.413 mmol) was added dropwise and with continuous stirring for 30 min at the same temperature. Subsequently, di-tert-butyl dicarbonate (Boc_2_O, 0.5268 g, 2.413 mmol) was added, the reaction mixture was allowed to reach room temperature and continued overnight with vigorous stirring (see Scheme 3). The progress of the reaction was monitored by TLC, and the resulting mixture was extracted with EtOAc (3 × 25 mL). The organic phase was washed with water and brine, dried over anhydrous Na_2_SO_4_ and concentrated to dryness. The residue obtained was purified by flash chromatography using a mobile phase of hexane/EtOAc (9:1), affording a 54% yield of compound **7** as a pearl yellow solid, R*_f_* = 0.63 (hexane/EtOAc 9:1), mp = 184.3 °C. HPLC purity = 95.50 %. ^1^H NMR (300 MHz, CDCl_3_) δ 7.88 (d, 1H, *J* = 1.5 Hz, H-9), 7.81 (dd, 1H, *J* = 8.4, 1.5 Hz, H-7), 7.61 (d, 1H, *J* = 8.8 Hz, H-6), 6.94 (s, 1H, H-7′), 6.29 (s, 1H, H-4′), 6.17 (m, 1H, H-1), 6.05 (d, 1H, *J* = 9.7 Hz, H-4), 5.85 (AB, 2H, H-2′), 5.64, (m, 1H, H-2), 3.85 (bd, 1H, *J* = 8.4 Hz, H-9b), 3.44 (q, 1H, *J* = 8.3 Hz, H-3a), 2.20 (dd, 1H, *J* = 16.3, 8.3 Hz, H-3 down), 1.79 (dd, 1H, *J* = 16.5, 6.8, Hz, H-3 up), 1.40 (s, 9H, O*t*Bu). ^13^C NMR (75 MHz, CDCl_3_) δ 165.6 (CO_2_H), 153.3 (C-carbamate), 147.0 (C-7′a), 146.9 (C-3′a), 141.4 (C-5a), 134.2 (C-9a), 132.2 (C-5′), 114.4 (C-6′), 81.7 (*C*-*t*Bu), 129.0 (C-9), 127.4 (C-7), 124.7 (C-6), 112.3 (C-7′), 107.9 (C-4′), 133.0 (C-1), 56.0 (C-4), 101.6 (C-2′), 131.8 (C-2), 43.5 (C-9b), 41.6 (C-3a), 34.9 (C-3), 28.2 (3Me-*t*Bu). HRMS (ESI) calculated for [C_25_H_24_BrNO_6_Na] + [M + Na]^+^: 536.0679; found: 536.0669.

### 3.9. Purity Analysis by HPLC

All HPLC analyses were performed using an Agilent 1260 infinity series liquid chromatograph (Agilent Technologies, Palo Alto, CA, USA) equipped with a quaternary pump delivery system (G1311B), robotic autosampler (G1316A), column thermostat (G1316A) and multiwavelength UV detector (G1315C), and the results were analyzed by OpenLab CDS EZChrom. The mobile phase with isocratic elution consisted of a mixture of (A) 0.2% acetic acid in water (*v*/*v*) at pH 3.0 and (B) CAN (acetonitrile) in a proportion of 40% A and 60% B. The system was ready for the next injection without further equilibration. The mobile phase was prepared fresh daily using deionized water that was filtered (0.22 μm) and degassed prior to use. The flow rate was 0.5 mL/min. The proposed method utilized a Zorbax SB-C18 column (5 μm, 4.6 × 150 mm) at a temperature of 25 °C. Acceptable purity was delimited when all compounds were confirmed by ≥95%.

### 3.10. Cell Culture

Four cell lines were used for this study. The human renal cell carcinoma cell line RCC4 was stably transfected with an empty vector, pcDNA3 (ECACC N-03112702; called RCC4-VA), which confers neomycin resistance, or with pcDNA3-VHL (ECACC N-0312703; called RCC4-VHL), which confers neomycin resistance and encodes the VHL tumor suppressor gene, pVHL, which is associated with resistance to chemotherapy. RCC4 plus the empty vector served as a negative control cell line to study the effects of pVHL expression by pcDNA3-VHL. The other two cell lines used in this study were the pancreatic cancer cell line MIA Paca-2 and the liver cancer cell line Hep G2. Cell culture medium, fetal bovine serum (FBS), *L*-glutamine, sodium pyruvate, minimum essential medium (MEM) nonessential amino acids, penicillin-streptomycin, triple express, phosphate-buffered saline (PBS) and bovine insulin were purchased from Life Technologies (Gibco, Invitrogen Corporation, CA, USA). MTT and methyl methanesulfonate (MMS) were purchased from Sigma Chemical Co. (St. Louis, MO, USA). Cells were maintained and plated with a robotic cell culture system, SelecT (TAP Biosystems). Hep G2 and MIA Paca-2 cells were obtained from the American Type Culture Collection (ATCC, Manassas, VA, USA). Hep G2 cells (human liver carcinoma, CCL-8065) were grown in ATCC-formulated Eagle’s MEM with 10% qualified FBS, 2 mM *L*-glutamine, 1 mM sodium pyruvate and 100 μM MEM nonessential amino acids. The MIA Paca-2 cell line (ATCC CRL-1420), a fibroblast primary pancreatic adenocarcinoma cell line that contains mutated K-RAS, P16 and P53 genes, was grown in Dulbecco’s modified Eagle’s medium supplemented with 10% fetal bovine serum, 2.5% fetal horse serum, 1% *L*-glutamine and 1% penicillin–streptomycin. RCC4-VA and RCC4-VHL cell lines were maintained in Dulbecco’s modified Eagle’s medium, supplemented with 10% FBS, 0.01% *L*-glutamine 200 mM, 0.01% penicillin–streptomycin (GIBCO) and 0.001% Geneticin G418 0.5 mg/mL (GIBCO). Cell cultures were grown at 37 °C in a humidified atmosphere of 5% CO_2_ and 95% air.

### 3.11. Antiproliferative Assays

The rate of MTT reduction is an indicator of the functional integrity of mitochondria and hence of cellular viability [64,65]. For this assay, cells were treated with compounds for 72 h. These cells were then seeded into 96-well plates at a density of 10^3^ cells/well for Hep G2, MIA Paca-2, RCC4-VA and RCC4-VHL. Three microliters of each pure compound were dispensed into 597 μL of fresh medium (see SM for purity statement). From this mixture, 200 μL was transferred to three different cell plates with the Biomek FX (Beckman Coulter) automated liquid handling system. MMS was used as a positive control and 0.5% DMSO was used as a negative control. The maximum concentration of DMSO was 0.5% to minimize any background solvent toxicity. Compounds were tested in triplicate with 10-point dilutions. After compounds and controls had been added, the plates were incubated at 37 °C in a 5% CO_2_ incubator for 72 h. After this time, the MTT solution was prepared at 5 mg/mL in PBS 1× and then diluted to 0.5 mg/mL in MEM without phenol red. The sample solution in each well was skimmed off, and 100 µL of MTT dye was added to each well. The plates were gently shaken and incubated for 3 h at 37 °C in a 5% CO_2_ incubator. The supernatant was removed, and 100 µL of 100% DMSO was added. The plates were gently shaken to solubilize the formed formazan. The absorbance was measured using a VictorTM multireader at a wavelength of 570 nm. One-way ANOVA test was performed to determine any significant difference between each compound treatment and the DMSO control; *p* > 0.05 was considered as statistically significant.

## 4. Conclusions

Bioinformatic tools, such as molecular docking and molecular dynamics simulations, provide relevant structural information, which allows the characterization of structural features that carry out molecular recognition. This offers a broad panorama in the design of selective drugs for the treatment of diseases. Under this context, we designed three compounds that showed notable inhibitory activity against GPER when tested in nonconventional cell models. According to the chemical data of the structural modifications of the GPER pharmacophore, we were able to identify the key entities (bromine atom and NH group of the piperidine ring) that could be functionalized to improve the activity of the compounds designed to target this receptor. We found a very good correlation of the in vitro experimental results with the in silico approximations on GPER. Our results suggest that GPER has the capacity to recognize different structural scaffolds of flat and aromatic character, specifically ligands with simple chemical modifications that improve the stereoelectronic properties of the pharmacophore. A comparison of the results obtained in this study with our previous structural insights demonstrates the great capability of GPER to bind ligands of different chemical structures, specifically ligands with simple chemical modifications over their pharmacophores. Nevertheless, additional assays, for example, binding assays and intracellular Ca^2+^ mobilization, will be crucial to corroborate our hypothesis. The appropriate use of rational in silico methods and tools could yield new compounds for the promising list of ligands to be used as potential drugs in cancer therapy.

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
