# Peer review of "Modifications on the Tetrahydroquinoline Scaffold Targeting a Phenylalanine Cluster on GPER as Antiproliferative Compounds against Renal, Liver and Pancreatic Cancer Cells"

_pharmaceuticals, 2021, doi:10.3390/ph14010049_

Round 1

Reviewer 1 Report

The authors have designed and synthesized three compounds for the growth inhibition of cancer cell lines. The synthetic procedures of this study are eye-catching and the experimental details were carefully conducted. However, there are several minor points in which improvements should be made to strengthen the manuscript.

  1. The author needs to perform the statistical evaluation (using statistical p-value) for Figure 4, 5, and 6.
  2. There is a lack of explanations of reduction of growth inhibition for #4 at the molar concentration of 6.3 uM.
  3. The addition of 1H NMR spectra of various compounds into the Supplementary Materials will be valuable for the readers.

Author Response

1. The author needs to perform the statistical evaluation (using statistical p-value) for Figure 4, 5, and 6.

Response: Thanks for your suggestion, the missing information about the statistical analysis performed to the MTT assay now is added in the text and highlighted in yellow both 2.11 Anti-proliferative assays section and Figures 4, 5 and 6.

2. There is a lack of explanations of reduction of growth inhibition for #4 at the molar concentration of 6.3 uM.

Response: The explanation of the possible mechanism was added in the corresponding section and is visible as text highlighted in yellow adding the phrase: … in this cell line is important to note the trend of 4 which shown a non-dose-dependent behavior (being evident at 6.3 µM concentration) may be due to intrinsic properties of the cell to adapt to increased concentrations of the tested compounds, in this case the cell could be increasing its expression of proteins involved into externalize the drugs (e.g. PGP ATPase) or another mechanism which confer to the cell the capability to not response to the treatments assayed, the above mentioned correlates with the result yielded by 7 assayed at the highest concentration, such result are slightly lower than the previous concentration (25 µM). This pharmacological phenomenon is named as biphasic pattern and  has been observed for other cancer cells (https://www.ncbi.nlm.nih.gov/pmc/articles/PMC4560104/)

3. The addition of 1H NMR spectra of various compounds into the Supplementary Materials will be valuable for the readers.

Response: Thanks for your suggestions, we have include all 1H as well as 13C NMR spectra at Supplementary Information.

Reviewer 2 Report

The manuscript entitled “Modification of bromine atom on the tetrahydroquinoline scaffold targeting a phenylalanine cluster on GPER as antiproliferative compounds against renal, liver and pancreatic cancer cells” presents results of docking studies done for 3 modified compounds that are believed to serve as potent inhibitors in the fight against cancer. In silico studies are focused on determining the binding posed and crucial interactions formed between these compounds and protein in generated GPER-ligand complexes. Additionally, the analysis of MD simulations and the estimation of the binding affinity of each of the compound to the intramembrane protein is done. In the opinion of this reviewer presented herein, studies are too weak to be published in Pharmaceuticals.

The main argument, that acts to the detriment of this work, stems from the difficulty of finding a link connecting theoretical and experimental research. While theoretical work focuses on one protein system, in experimental work much more variables are introduced (a full cancer cell), thus perturbing the possibility to obtain a direct connection between obtained results. Moreover, estimated values of free energy binding presented in Table 1 and Table 3 do not correlate in any way with experimentally measured values of IC50. Finally,  values of IC50 obtained in selected cancer cell lines suggest the existence of meaningful differences in ligand affinities in each of them, and thus no clear conclusion can be driven from these studies until more analysis will not be done.   

Minor questions and comments:

1.          All missing FF parameters generated for 3 used in docking and MD simulations compound should be provided in SI document.

2.          How the exact stereo form of ligand used in docking studies was estimated? Was it taken from the compound library?

3.          do any experimental evidence exist that would support estimated in the previous docking studies location of the inhibitors? What about blind docking test did AutoDock was able to find only this binding site or more possible binding cavities can be defined?

4.          On page 3 line 112 it is written: “the protein binding site with the 112 following grid box parameters: XYZ of 60 Å3 over the Cα atom of the N310 residue”. What are exact sizes of the dimensions of the box 60x60x60 or x,y,z that in total gives volume od the box of 60 Å3. This should be clarified.

5.          How the protonation of the titratable residues present in the GPER structure was established. Is any shift in pKa observed for these residues?

6.          What exact variables were used in cluster analysis that allowed to differentiate different conformer stages of the protein along MD simulations?

Author Response

1.          All missing FF parameters generated for 3 used in docking and MD simulations compound should be provided in SI document.

Response: Thanks for you suggestion, this information has been added as supplementary Information and is attached here as a .pdf file (forcefield-parameters.pdf) as a first review.

2.          How the exact stereo form of ligand used in docking studies was estimated? Was it taken from the compound library?

Response: According to the compounds yielded by the chemical reaction (three-component aza-Diels Alder (Povarov) it was expected for the 3 compounds to obtain the endo diastereomer excess, thus, all the ligands employed in the in silico evaluations were constructed on the basis of its endo diasteromer forms. Its important to note that, Burai et. al. previously reports the synthesis conditions to perform the obtention of the tetrahydroquinoline scaffold with increased diastereoselectivity (https://pubmed.ncbi.nlm.nih.gov/20401403/). 

3.          do any experimental evidence exist that would support estimated in the previous docking studies location of the inhibitors? What about blind docking test did AutoDock was able to find only this binding site or more possible binding cavities can be defined?

Response: There is a lack of knowledge about crystal structures of GPCRs bound to its cognate ligands due to the intrinsically disorder inherent to the nature protein, however, our previous studies reported by Mendez-Luna et. al. (https://pubmed.ncbi.nlm.nih.gov/25587872/, 2014), for GPER, combining molecular docking to obtain the binding pose (output docking topology) of several known GPER-ligands tested in biological assays explored under blind docking procedure, were analyzed under two main criteria, first, the most exergonic binding energy and second, the most populated docking conformation, always correlated both criteria for the ligands tested. Besides that, the output docking topologies of the ligands tested, were summited to molecular dynamics simulations in order to analyze ligands stabilization, H bond networks and other parameters in which, in 2016 Mendez-Luna et. al. (https://pubmed.ncbi.nlm.nih.gov/26772481/) described the potential behavior of both synthetic agonist (G1) and an antagonist (G15) for GPER at atomistic level, describing in the two mentioned research that for GPER is possible to describe additional cavities but being the most important for the molecular recognition the Phenylalanine cluster formed for Phe 206, 208 and 314.    

4.          On page 3 line 112 it is written: “the protein binding site with the 112 following grid box parameters: XYZ of 60 Å3 over the Cα atom of the N310 residue”. What are exact sizes of the dimensions of the box 60x60x60 or x,y,z that in total gives volume od the box of 60 Å3. This should be clarified.

Response: Thanks for you suggestion, we denotated 60 Å3 in each coordinate to allow that, according to the length of the parameter, all the GPER-residues involved in molecular recognition were able to be reached for the ligands here tested, these grid-box measurements are according to the reported elsewhere for the focused docking procedure. It was follow according to our previously in silico studies (Mendez-Luna et. al. (2014) and Mendez-Luna et. Al 2016)

5.          How the protonation of the titratable residues present in the GPER structure was established. Is any shift in pKa observed for these residues?

Response: Prior to MD simulations, GPER 3D model was properly protonated in the specific residues that were required with PDB2PQR which prepares structures for further calculations by reconstructing missing atoms, adding hydrogens, assigning atomic charges and radii from specified force fields (https://www.cgl.ucsf.edu/chimera/docs/ContributedSoftware/apbs/pdb2pqr.html) and re-analyzed with the SANDER module included in AMBER package for MD simulations, setting for all the simulations a pH of 7.4 for all the titratable residues, therefore, any shift was observed in the simulations.       

6.          What exact variables were used in cluster analysis that allowed to differentiate different conformer stages of the protein along MD simulations?

Response: The clustering analysis was performed considering a cut-off of 0.25 nm to identify the most populated conformer which was employed to describe the most predominant interactions present in the receptor-ligand complex.

Reviewer 3 Report

This work by Mendez-Luna et al is a joint investigation combining computational tools and experimental assays in order to design potential drug candidates inhibiting the GPER. The authors designed three promising candidate compounds and carried out molecular docking, molecular dynamics simulations and measured the inhibition strength on 4 cancer cell lines.The research was designed well and a few minor issues need to be addressed.

1) The title talks about the bromine atom of the drug candidates, however, among the three drug candidates (4,5,7) designed by the authors, only the last one (7) is related to Br atom. Therefore, the title might need to be adjusted to better suit the content of this manuscript.

2) The abstract contains some redundant information. For example, the "four cancer cell lines" appeared twice. And only three cancer cell line names were listed. Please improve the abstract.

3) In section 2.1, the authors wrote "Gaussian 9.0", which is wrong. 

4) In section 2.4 line 156, the "GROMOS method" is not correct. Please double check.

5) In Table 1, the "no bonded interactions" should be "non-bonded interactions".

6) The author did not show the figure for the results of molecular docking. How does this result match with the MDS result? 

7) The ordering of three compounds in different results is inconsistent. Sometimes it is 7-4-5 and sometimes it is 7-5-4. Please improve.

Author Response

1) The title talks about the bromine atom of the drug candidates, however, among the three drug candidates (4,5,7) designed by the authors, only the last one (7) is related to Br atom. Therefore, the title might need to be adjusted to better suit the content of this manuscript.

Response: Thank you for your kind suggestion, the title was modified removing the referring to the Bromine atom to add a better understanding of the work, it is now appreciated in the text as highlighted in yellow. It is now: “Modifications on the tetrahydroquinoline scaffold targeting a phenylalanine cluster on GPER as antiproliferative compounds against renal, liver and pancreatic cancer cells”.

2) The abstract contains some redundant information. For example, the "four cancer cell lines" appeared twice. And only three cancer cell line names were listed. Please improve the abstract.

Response: Thanks for you suggestion, the abstract now is modified and it is appreciated in the text as highlighted in yellow.

3) In section 2.1, the authors wrote "Gaussian 9.0", which is wrong. 

Response: Thanks for you annotation, the correct form to name the software now is corrected and you can see it in the text highlighted in yellow.

4) In section 2.4 line 156, the "GROMOS method" is not correct. Please double check.

Response: Thanks for you annotation, the mistake was corrected (please check line 154 - 155, page 4)

5) In Table 1, the "no bonded interactions" should be "non-bonded interactions".

Response: Thanks for you annotation, the modification now is done, and is visible in the text and highlighted in yellow.

6) The author did not show the figure for the results of molecular docking. How does this result match with the MDS result? 

Response: The figure is now uploaded as SI for a better understanding of the work, but according to this, just few modifications in the binding pose were yielded by MD simulations for the tested ligands, therefore we can conclude that the docking results were in-line with the data yielded by MD simulations.

7) The ordering of three compounds in different results is inconsistent. Sometimes it is 7-4-5 and sometimes it is 7-5-4. Please improve.

Response: Thanks for you suggestion, the modification was done in each part of the document in which apply the correct order was necessary and can be read in the text, Figures and Tables as highlighted in yellow.

Round 2

Reviewer 2 Report

Unfortunately, authors have provided answers to just minor comments, not mentioning the main problems indicated in the previous review. There was no effort made to defend, key in the opinion of the reviewer, problems found in presented studies. (1) Theoretical work focuses on one protein system, in experimental work much more variables are introduced (a full cancer cell), thus perturbing the possibility to obtain a direct connection between obtained results. (2) values of IC50 obtained in selected cancer cell lines suggest the existence of meaningful differences in ligand affinities in each of them, and thus no clear conclusion can be driven from these studies. Therefore, this reviewer has no choice but to uphold earlier decisions and reject this MS.

 Here more comments about the answer provided by authors are added:

The answer “we denoted 60 Å3 in each coordinate to allow that, according to the length of the parameter” provided by authors is clear. But still, the notation used in the revised version of MS is wrong (XYZ of 60 Å3). Or it should be xyz of 60x60x60 Å3 or box of 216 000 Å3 volume. This should be corrected.

Again, answer about deciding about protonation states of titratable residues is correct, but the description should be added to the text. This information is crucial in order to be able to reproduce the results obtained in these studies.

The authors did not provide a sufficient answer for the previous question “What exact variables were used in cluster analysis that allowed to differentiate different conformer stages of the protein from MD simulations?” i.e. which interactions were chosen and the range of values to characterized each of conformer? How many clusters were identified and what is their characteristic?